# Translocation of Hydrophobic Polyelectrolytes under Electrical Field: Molecular Dynamics Study

**DOI:** 10.3390/polym15112550

**Published:** 2023-05-31

**Authors:** Seowon Kim, Nam-Kyung Lee, Min-Kyung Chae, Albert Johner, Jeong-Man Park

**Affiliations:** 1Department of Physics and Astronomy, Sejong University, Seoul 05006, Republic of Korea; 2National Institute for Mathematical Sciences, Daejeon 34047, Republic of Korea; 3Institut Charles Sadron CNRS-Unistra, 6 Rue Boussingault, CEDEX, 67083 Strasbourg, France; albert.johner@ics-cnrs.unistra.fr; 4Department of Physics, the Catholic University of Korea, Bucheon 14662, Republic of Korea

**Keywords:** translocation, polyelectrolytes, Fokker–Planck equation, molecular dynamics simulation

## Abstract

We studied the translocation of polyelectrolyte (PE) chains driven by an electric field through a pore by means of molecular dynamics simulations of a coarse-grained HP model mimicking high salt conditions. Charged monomers were considered as polar (P) and neutral monomers as hydrophobic (H). We considered PE sequences that had equally spaced charges along the hydrophobic backbone. Hydrophobic PEs were in the globular form in which H-type and P-type monomers were partially segregated and they unfolded in order to translocate through the narrow channel under the electric field. We provided a quantitative comprehensive study of the interplay between translocation through a realistic pore and globule unraveling. By means of molecular dynamics simulations, incorporating realistic force fields inside the channel, we investigated the translocation dynamics of PEs at various solvent conditions. Starting from the captured conformations, we obtained distributions of waiting times and drift times at various solvent conditions. The shortest translocation time was observed for the slightly poor solvent. The minimum was rather shallow, and the translocation time was almost constant for medium hydrophobicity. The dynamics were controlled not only by the friction of the channel, but also by the internal friction related to the uncoiling of the heterogeneous globule. The latter can be rationalized by slow monomer relaxation in the dense phase. The results were compared with those from a simplified Fokker–Planck equation for the position of the head monomer.

## 1. Introduction

In nature, charged polymers such as proteins or RNAs are imported into, or exported from, the cell nucleus by translocation through a nuclear pore [1]. These processes involve helper molecules (importins/exportins). Translocation through a single pore has also been implemented in the laboratory [2], which was mainly performed to study DNA sequences carrying negative charges only. In laboratory setups, translocation is usually driven by an electric field, and the applied trans-pore electric potential, around 200 mV, is much larger than the cell transmembrane potential of ∼50–70 mV.

The translocation of polyelectrolytes (PEs) [3,4,5] is ensured by an external electric field with an electric potential drop that is localized across the pore [6]. A long open PE chain experiences entropic confinement and a free energy penalty upon insertion in the pore, which acts against translocation. In moderate salt and weakly poor solvent conditions, a PE forms partially globular structures of the pearl-necklace type, due to the interplay between electrostatic repulsion and cohesive surface energy [7,8]. The presence of open tails helps the PE to be captured by the pore [9]. However, the globular part should unfold while engaging into the pore. Nucleating a small globule on the *trans* side can entail a (secondary) free energy barrier against translocation if the solvent is poor enough. High salt conditions are typical in vitro, where the electric current of small ions is essential. In high salt conditions, the electrostatic repulsion is screened, and the PE forms a single globule. The neutral monomers are hydrophobic (H), while charged ones are polar/hydrophilic (P). There are H/P composition profiles inside the globule. However, due to the high salt, electrostatics plays no direct role in the PE structure. The energetic aspect of the unwinding of heteropolymers by external force has been studied using Monte Carlo simulation, and it was found that the hydrophobic part (core) was more resilient against unfolding [10].

In this work, we focus on the influence of the hydrophobicity and the heterogeneous structure of PE globules on PE translocation. This dynamic process was studied using molecular dynamics simulations. We assumed an overall high salt condition where electrostatic interactions between monomers were mainly screened. We only distinguished the hydrophobicity of the monomers and assumed that each monomer was either charged hydrophilic (P) or neutral hydrophobic (H). The so-called HP model [11,12,13] captured the corresponding physics.

In order to consider the channel properties, we modeled the translocation of the PEs through the α-hemolysin pore by applying a realistic force field extracted from the channel potential along the pore [14,15]. α-hemolysin is representative of the class of asymmetric pores and is widely used for the in vitro study of translocation. α-hemolysin has a (positively) charged *cis* protrusion, which favors the presence of (negatively) charged biopolymer sequences in the vicinity of the pore entry. The *trans* edge is negatively charged. Two regions, the *cis* vestibule and a *trans* channel stem, are separated by a narrow constriction, about 3 nm long and 1 nm wide, which is essentially polar. The α-hemolysin channel has an electrostatic structure that influences the preferential translocation of negatively charged ions. In an early Monte Carlo simulation study [15], we incorporated realistic potential profiles of the α-hemolysin pore in a discrete manner to investigate the sequence dependence of the translocation of polyampholytes. In this study, we utilized the continuous force field of α-hemolysin to incorporate molecular dynamics simulations. The translocation dynamics of negatively charged *open* polymers through the α-hemolysin have been addressed by Muthukumar et al. [16,17,18]. Some simulation studies for open PE translocation incorporating electrostatic interactions are available for a generic pore with an explicit salt [5,19]. There are only a few works on the translocation of homopolymers in poor solvent conditions [20,21]. To our knowledge, the translocation of globular heteropolymer PEs, such as globular proteins abundant in nature whose conformations are partially denatured or possibly in the wet globule state [22], has not been addressed.

We performed molecular dynamics simulations of PEs to obtain the statistics of translocation behaviors for various solvent conditions and for various chain lengths. Based on a simplified theoretical model in which electrical and solvent effects are incorporated into the potential energy for PE translocation, we rationalized the time scales obtained from simulations.

## 2. Molecular Dynamics Simulation

### Simulation Model

In the molecular dynamics simulations, we considered PE chains crossing a wall through a pore. A PE chain consists of charged monomers and neutral monomers, both of which have a size of σ. Every third monomer carries a (negative) unit (elementary) charge −1. These P-type hydrophilic monomers were labeled as 1. Other monomers are electrically neutral without a charge. These H-type hydrophobic monomers were labeled as 0. Our simulations considered chains of various lengths (100)${}_{n}$, (010)${}_{n}$, and (001)${}_{n}$ in which (100), (010), and (001) were repeated *n* times, respectively.

We modeled the PE using a bead-spring chain of charged Lennard–Jones (LJ) particles consisting of *N* beads each with a diameter σ. The interaction between two particles was modeled by an LJ potential: $U_{\mathrm{LJ}}\left(r\right)$ = $\epsilon_{\mathrm{LJ}}\left[{\left(\sigma /r\right)}^{12}-2{\left(\sigma /r\right)}^{6}-{\left(\sigma /r_{c}\right)}^{12}+2{\left(\sigma /r_{c}\right)}^{6}\right]$. Here, $\epsilon_{\mathrm{LJ}}$ represents the strength of the LJ potential; *r* denotes the center-to-center distance between two interacting particles; and $r_{c}$ is the cut-off distance. The excluded volume interactions were modeled by the Weeks–Chandler–Andersen (WCA) potential.

For interactions between neutral monomers, we used the LJ potential without a cut-off length. Different values of $\epsilon_{\mathrm{LJ}}=\epsilon_{\mathrm{HH}}$ were chosen to take into account various values of hydrophobicity. We assumed that the electrostatic interactions outside of the channel were screened, but charged monomers remained hydrophilic. Hence, we set the interaction parameters $\epsilon_{\mathrm{PP}}$ between two charged monomers to be 0.6 $k_{B}T$ with a cut-off distance $r_{c}$ = $2^{1/6}\sigma$ [12]. These interactions were considered to be purely repulsive. The same repulsive interactions were assumed between a neutral monomer and a charged monomer $\epsilon_{\mathrm{HP}}=\epsilon_{\mathrm{PP}}$. Charged monomers were assumed to only be under the influence of the external electric field inside the channel; hence, they were driven from the *cis* to the *trans* side.

In order to vary the solvent conditions, we set the value of the interaction parameter $\epsilon_{\mathrm{HH}}$ between neutral monomers 0.3, 0.5, 0.7, 0.9, and 1.1 $k_{B}T$ so that the polymers would be near the theta or poor solvent condition. The chain connectivity was ensured by the finite extension nonlinear elastic (FENE) potential between two consecutive beads, $U_{\mathrm{FENE}}\left(r\right)=-0.5\;kr_{0}^{2}\;\ln\left[1-{\left(r/r_{0}\right)}^{2}\right]$, where the spring constant is taken as *k* = 30 $k_{B}T/\sigma^{2}$ and the maximum bond length as $r_{0}$ = 1.5σ [23].

A bilayer membrane was modeled as a double wall of thickness l=5σ. Considering that α-hemolysin pore is 10 nm long, 1σ is about 2nm in real units. The wall divides the space into two subspaces: the *cis* and *trans* regions. For simulations, the common steric influence was handled by including the repulsive forces caused by the particles that made up the wall. Our simulation was set in infinite space. We put finite size membrane in a separate cis side and trans side. Each wall of membrane spanned lateral areas of 20σ×20σ for *N* = 30 or 40σ×40σ for *N* = 90, which is large enough to ensure that the chain does not cross the membrane from the outer periphery in a good solvent condition. Figure 1 illustrates the wall, which had the lateral dimensions of 20σ×20σ. The pore was a 2σ×2σ hollow at the center of the membrane. As we were mainly interested in the behaviors of polymers, we assumed that beads constituting the wall were not mobile. The Lennard–Jonnes parameters between the mobile monomers and immobile beads were set to be $\epsilon_{\mathrm{wm}}$ = 1.0 $k_{B}T$ with a cut-off distance of $r_{c}$ = $2^{1/6}\sigma$ so that the interactions would be purely repulsive.

A pore perforated the membrane along *z*-direction.

The full scale simulation, including charge structure and the polarization of materials due to the electric field, would be a formidable task to execute. We first extracted the electrostatic part of the free energies $U_{el}$, which was applied the to charged particles inside of the channel in accordance with Ref. [14] (See Figure 2). A driving electric field was set across the pore (−1σ<z<5σ), and the potential difference between the *cis* and *trans* regions was ∼5.8 $k_{B}T$ per elementary charge, which corresponds to 150 meV at room temperature. Hence, the magnitude of force, on average, acting on the charged monomer would be ∼$1\;k_{B}T/\sigma$, which corresponds to ∼2 pN in real unit. The net electric force acting on the chain depends on the number of charges and their positions inside the channel. The electrostatic contribution to the free energy isthe opposite for cations and anions, and the odd part of the free energy corresponding to the electrostatic energy vanishes when averaging over. We took the average in order to get the remaining even part of the free energy with respect to the charge, and we subtracted the average to get the odd part, which was assimilated to the electrostatic contribution of free energy $U_{el}$. It was straightforward to get the force by taking the derivative from the odd part of the free energy for small ions: $-\frac{dU_{el}}{dz}$. The corresponding force field of the α-hemolysin pore of electrostatic origin is shown in Figure 2.

In order to describe the motion of beads, we integrated the following equation with the total energy *U*:$$M\frac{dv(t)}{dt}=-\zeta \frac{\partial r_{i}}{\partial t}-\frac{dU}{dr_{i}}+f^{R}\left(t\right),$$
where ζ is the frictional coefficient, and *M* is the mass of the bead. The Gaussian random force **f**${}^{R}$ has a zero average $\langle f^{R}\left(t\right)\rangle =0$ and correlations $\langle f^{R}\left(t\right)\cdot f^{R}\left(t^{\prime }\right)\rangle =6k_{B}T\zeta \delta \left(t-t^{\prime }\right)$. We integrated Newton’s equations of motion using the velocity Verlet algorithm with an integration time step δt = 0.005 $t_{0}$, where $t_{0}=\sigma {\left(1/k_{B}T\right)}^{1/2}$ is the characteristic time scale with bead mass *M* = 1. A Langevin thermostat with the damping constant 1.0 $t_{0}^{-1}$ was used to keep the system at the fixed temperature *T* = 1.0.

Since only a very small percentage of attempts was successfully engaging, we set the head monomer at the *trans* side channel edge (*z* = 0 in Figure 1) and equilibrated the chain before the measurement of translocation time. We first performed runs over time ∼$N^{2}t_{0}$ for the chain to relax to its equilibrium distribution, with the head monomer kept at the trans side wall (*z* = 0). After equilibration, we ran until the chain was translocated or rejected. The successful translocation means that the tail of the chain leaves the pore completely to the *trans* side (z≤−1σ). If the head of the chain is retracted to the *cis* side boundary (out of the pore), z≥5σ, it is considered to be a rejection. Once the chain is rejected, it is not allowed for re-entry. Any attempt to enter the pore should overcome the entropic free energy barrier. It is treated as a new attempt.

The probability of being captured would depend on the channel structure. Here, we may assume that the PE is captured when the head monomer is located inside the channel ∼2σ away from the trans-side wall (Figure 1). Prior to the simulation study of the translocation time of a (hydrophobic) PE, we obtained the probability that the head of the captured polymer reached the *trans* side wall, which was the starting position of the simulations. The probabilities that the head of the chain of *N* = 30 would reach the *trans* side wall (*z* = 0) starting at the captured position (z=2σ) were 0.17, 0.13, 0.07, 0.03, and 0.01 with average times of 24–29 (29, 28, 25, 24, and 29) and standard deviations of 15–22 for $\epsilon_{\mathrm{HH}}$ = 0.3, 0.5, 0.7, 0.9, and 1.1, respectively. Below, we report the translocation time starting at the *trans* side wall.

## 3. Results

### 3.1. Translocation Times and Their Dispersions: (100)${}_{n}$ Sequences

We measured the number of monomers in the *trans* side $n_{t}$ to identify translocation times for various solvent conditions characterized by $\epsilon_{\mathrm{HH}}$. If a monomer entered the *trans* region (z≤−1σ), it was counted as a translocated monomer. We show the translocation times $t_{t}$ for chain length *N* = 30 and *N* = 90 in Table 1 and Figure 3.

The waiting time $t_{w}$ is the MD time required for the number of *trans* side monomers to be $n_{t}$ = 5. The chain does not retreat back once the number of translocated monomers $n_{t}$ reaches five. This cut off point $n_{t}$ has been found to be nearly independent of the chain length *N*. This is consistent with the fact that the retreating force mainly depends on the local structure of the globule and is almost independent of the chain length. In poor solvent conditions, $n_{t}$ = 5 suffices to ensure nucleation in the *trans* side. This condition does not depend on the remaining chain length in the *cis* side. In good or intermediate solvent conditions, the entropic free energy of the chain dangled in the *cis* side only grows with chain length logarithmically. Hence, the free energy barrier to enter the pore has very weak *N* dependence, so the cut-off point remains nearly constant. In our model, $n_{t}$ is a discrete variable, and every third monomer carries a charge. This periodicity suppresses the weak variation in the cut-off point that is expected in a continuous model.

After $t_{w}$, the time until the tail of the chain leaves the channel is the drift time $t_{d}$. The total translocation time $t_{t}$ is the sum of $t_{w}$ and $t_{d}$ (See Figure 4). The waiting time $t_{w}$ shows stronger solvent condition dependence than $t_{d}$, which is mainly determined by the chain length. The overall translocation time $t_{t}$ increases with increasing $\epsilon_{\mathrm{HH}}$ values for $\epsilon_{\mathrm{HH}}\geq 0.5$, together with relative standard deviation. In the driven regime, $n_{t}\left(t\right)$ increases with time (step-wise) linearly with some fluctuations of ∼±2. For very poor solvent conditions, $\epsilon_{\mathrm{HH}}\geq 0.9$, the process accelerates at late times, which implies the formation of stable globules in the *trans* side. Some movie files are available in Appendix A. Consistent with the results restricted to homopolymers [20], we found that the translocation time at good solvent (e.g., $\epsilon_{\mathrm{HH}}$ = 0.1) or theta solvent (e.g., $\epsilon_{\mathrm{HH}}$ = 0.3) conditions was somewhat longer than that in weakly poor solvent conditions (e.g., $\epsilon_{\mathrm{HH}}$ = 0.5) because the PE was not globular yet and did not need to be uncoiled. The PE at $\epsilon_{\mathrm{HH}}$ = 0.5 was slightly more compact than at $\epsilon_{\mathrm{HH}}$ = 0.3, and the entropic force against translocation was smaller. In contrast to the results reported for homopolymers [20], the minimum in the translocation time at a weakly poor solvent condition was very shallow, and, for moderate hydrophobicity, the translocation time appeared almost independent of $\epsilon_{\mathrm{HH}}$. After the shallow minimum, the translocation time markedly increased with hydrophobicity. Given that we considered a regular PE, the dispersion in translocation times remained moderate (See Figure 3d).

The entropic penalty is against the engagement of the PE toward the narrow channel. In good solvent conditions, the electrostatic driving force $f_{el}$ should be balanced by the entropic force at the start. At the steady state, the net force is balanced by friction, $f=\zeta \frac{dn_{t}}{dt}$. As translocation proceeds, more monomers are accumulated in the *trans* side. Then, entropic force is favorable for translocation, and the drift becomes faster. The drift speed is almost constant at $\epsilon_{\mathrm{HH}}=0.5.$ In poor solvent conditions, the globule should unfold in order to engage through the narrow channel. The friction involved in passing through the corrugated channel potential adds up with the internal globule friction upon unwinding. As more and more monomers translocate, the surface tension becomes favorable for translocation and further promotes drift. Upon analyzing the translocation process (See Figure 5a), we distinguished two regimes: (1) the waiting period and (2) the driven period. The captured chain was waiting for further uncoiling or to overcome entropic force. The waiting period ended when the translocated length reached the sufficient length $n_{t}^{*}$(≈ 5).

As shown in Figure 3 and Table 1, for the given *N*, $t_{d}$ moderately increased with $\epsilon_{\mathrm{HH}}$. The effective friction $\zeta^{\mathrm{eff}}$ also increased moderately with increasing $\epsilon_{\mathrm{HH}}$ (Table 2). The simulation data in Figure 5b is the average of 100 trials after resetting $t_{w}$ to *t* = 0. The effective friction $\zeta^{\mathrm{eff}}$ reflects the channel friction and internal globule friction, depending on the density of the globular structure. In an early regime of the drift, $n_{t}\left(t\right)$ grows with time linearly. We extracted the effective friction $\zeta^{\mathrm{eff}}$ from the slope via the relation $\zeta^{\mathrm{eff}}∼f_{e}/\left(dn_{t}\left(t\right)/dt\right)$ and assumed that the driving force $f_{e}$ was almost constant. This result for $\zeta^{\mathrm{eff}}$ is summarized in Table 2.

The slope shows the time dependence of the translocation speed (Figure 5b). The flat speed in the inset captures the drift regime. For $n_{t}\left(t\right)>N/2$, the slope $dn_{t}\left(t\right)/dt$ changed as the *trans* globule promoted the translocation process at later times, and the effective friction was reduced accordingly. This effect was more evident for larger $\epsilon_{\mathrm{HH}}$. The process accelerated as the imbalance of surface tension contributed as the driving force. The dependence of the translocation speed on $n_{t}$ could be interpreted as the change in the effective friction as reported for polymers ejected out from a cavity [24,25].

In adopting the HP model of globule, we tried to obtain the friction constant ζ for the given solvent quality. We assumed that the free energy of the HP globules on the *cis* and *trans* sides consisted of surface energy and potential energy under the electric field across the channel. With $n_{t}$ monomers in the *trans* side, the free energy accounting for surface tension and external electric potential is defined as follows:(1)$$F=k{\left(1-1/p\right)}^{2/3}\left\{n_{t}^{2/3}+{\left(N-n_{t}-l\right)}^{2/3}\right\}-\frac{n_{t}}{p}\left|U_{e}\right|,$$
where *p* is the periodicity of the charge sequence. Because the channel is filled, it does not contribute to the free energy variation, and is filled up to fluctuations in monomer and charge content of the channel. In the driven regime, we solved the simplified kinetic equation for $n_{t}\left(t\right)$:(2)$$k^{\prime }\left\{-{\left(n_{t}\right)}^{-1/3}+{\left(N-n_{t}-l\right)}^{-1/3}\right\}+\frac{1}{p}\left|U_{e}\right|=\zeta \frac{dn_{t}}{dt}.$$

We compared the $n_{t}\left(t\right)$ obtained from simulation and from the numerical solution of the Equation (Equation 2) for $t>t_{w}$ and extracted the prefactor $k^{\prime }=\frac{2}{3}k{\left(1-1/p\right)}^{-1/3}$ and ζ (See Figure 5). Note that, with *p* =3, it happened to be that $k^{\prime }$ was equal to the prefactor of the free energy term in Equation (Equation 1), $k^{\prime }=k{\left(1-1/p\right)}^{2/3}$. The fitting results for $k^{\prime }$ and ζ are summarized in Table 2. The fitted values of ζ were somewhat smaller than $\zeta^{\mathrm{eff}}$ from the early driven regime. This reflects the nucleation of the trans globule accelerating the translocation. Consistent with the fact that the translocation time $t_{t}$ and the drift time $t_{d}$ had minimums at $\epsilon_{\mathrm{HH}}$ = 0.5, both ζ and $\zeta^{\mathrm{eff}}$ were smallest for $\epsilon_{\mathrm{HH}}$ = 0.5.

We further investigated the influence of the chain length *N* on translocation. The measured values are shown in Figure 3b, and the values at $\epsilon_{\mathrm{HH}}$ = 0.9 are summarized in Table 3. The drift time $t_{d}$ increased approximately linearly with *N*, and the magnitude of the effective friction coefficient also tended to increase. The waiting time $t_{w}$ showed moderate dependence on the chain length *N*. The standard deviation remained as large as the average values of $t_{w}$ (Figure 3d).

The effective friction $\zeta^{\mathrm{eff}}$ is related to the relaxation of uncoiling the globule [26,27,28,29,30,31]. We checked the globule structures for *N* = 30 and *N* = 90 for various values of $\epsilon_{\mathrm{HH}}$. The monomer density correlation function g(r) and density ρ of *cis* side globules are shown in Appendix B.

As the monomer relaxation time $t_{0}$ had a larger value in a polymer melt with a high density [32,33], the unit time $t_{0}$ was larger in polymer globules with higher density, and, thus, the effective friction coefficient increased with the density of the globules (See Appendix B). Although the drift regime is well defined on average for most of $\epsilon_{\mathrm{HH}}$, each trajectory of translocation can differ significantly from the average, especially at poor solvent conditions. In Figure 6, we plotted several typical trajectories of PEs of *N* = 90 at $\epsilon_{\mathrm{HH}}$ = 0.9. The slow progression of translocation in the drift regime, depicted as plateaus in $n_{t}\left(t\right)$ trajectories, can be attributed to the slow monomer relaxation in the dense core of the globule as opposed to the P-rich corona (see also Ref. [10]).

Note that the equilibrated initial *cis* side globule has a well optimized free energy and is tougher to uncoil than the fresh-folded *trans* side globule. Therefore, the friction upon the folding of the *trans* side globule is lower than the friction upon extraction from the equilibrated *cis* side globule. Because the local structure and interface (a few monomers in size) relax very fast and the free energy relaxes very fast after/upon folding, the measured density profiles of H- and P-type monomers in the *trans* side globules (not shown) are alike with those in the equilibrated *cis* side globule (shown in Appendix B
Figure A1). The full intermixing inside the globule between older and newer parts is slower and mainly affects the unwinding friction of the *trans* globule (in the case of retraction). The effect of aging on friction would be much stronger for larger entangled globules [26,27,30]. Figure 7 demonstrates mixing degrees of monomers in the *trans* and *cis* globules. Monomers are colored as blue to red along the contour from head to tail. The mixing of colors is less frequent in the *trans* side.

Table 4 summarizes the sequence dependences of translocation times for three sequences: (100)${}_{10}$, (010)${}_{10}$, and (001)${}_{10}$. When the chain started with favorable charge as a head monomer, the capture probability was high. At a good solvent condition of $\epsilon_{\mathrm{HH}}$ = 0.3, the translocation probability of (100)${}_{10}$ was ∼1, but (001)${}_{10}$ translocated with a probability of 0.29. At a poor solvent condition $\epsilon_{\mathrm{HH}}$ = 0.9, (100)${}_{10}$ had a long waiting time but still had a good success rate of 0.94. In contrast, most of the trials of (001)${}_{10}$ were rejected, and successful ones had relatively short translocation times.

### 3.2. Free Energy Profiles and the Solution of Fokker-Planck Equation

In order to rationalize the solvent-dependent behavior of translocation times, we solved a Fokker–Planck equation for the position of the head monomer. The profile of the potential energy *U* can be obtained as a function of the number of monomers, $m=n_{t}+n_{p}$, residing in the *trans* region ($n_{t}$) and pore ($n_{p}$) (i.e., z≤3σ). There are mainly three contributions: (a) the channel entropic effect $F_{c}$, (b) electrostatic potential energy $F_{e}$, and (c) surface energy $F_{s}$; the latter is relevant in the poor solvent conditions. The early translocation process m<5 was mainly controlled by $F_{c}$. The reference energy value was taken as the energy in the *cis* side. As a monomer engaged into the channel, the free energy increased until the monomer reached the *trans*-side channel end, which was mainly due to the entropic penalty, even in the presence of the electric field favorable for translocation. The electrostatic contribution to the free energy decreased by 5.8 $k_{B}T$ per translocated charge. We set the electrostatic potential energy as $F_{e}$ = −1.9 *m*, assuming that 1/3 of monomers were charged (i.e., *p* = 3). The surface energy is calculated as:(3)$$F_{s}=\left\{\begin{array}{cc} k^{\prime }{\left(N-m\right)}^{2/3}, & (m\leq l),\\ k^{\prime }\left\{{\left(N-m\right)}^{2/3}+{\left(m-l\right)}^{2/3}\right\}, & (m>l). \end{array}\right.$$

In order to obtain a more precise form of the free energy over a wider range of translocation states, especially to cover small *m* regions while taking into account the influence of the entropic effect $F_{c}$, we exploited the following simulation to compute the success rate. We counted successful attempts of arrivals at a specific position, *m* = *b*. As shown in detail in the Appendix C, the success rate $\pi_{b}\left(x\right)$ is related to the potential energy *U* by the following equation:(4)$$\pi_{b}\left(x\right)=\frac{\int_{a}^{x}e^{U(y)}dy}{\int_{a}^{b}e^{U(y)}dy},$$
with the fixed (absorbing) boundaries at a,b and the starting point at *x*. The differential of the success rate is $\frac{d\pi_{b}\left(x\right)}{dx}∼e^{U(x)}$, and the logarithm of the differential of the success rate yields ∼U(x). In order to get the free energy profile, we evaluates the success rate for various values of *m* in a discrete manner. (Figure 8a) The translocation processes were repeated for $∼10^{4}$–$10^{5}$ times, with the PEs equilibrated with fixed number *m*. After taking the logarithm of the difference in success rate, $\pi_{b}\left(m+1\right)-\pi_{b}\left(m\right)$, we obtained the potential energy $U(m+\frac{1}{2})$. This approach is generally valid and operational until the top of the (last) barrier $m^{†}$ is reached and the measured success rate saturates to 100%. Since the potential energy is mainly given by $F_{e}$ and $F_{s}$ in the following drift regime, we used the free energy expression $F=F_{e}+F_{s}$ starting from $m^{†}$.

Based on the discrete points obtained, we constructed the potential profiles U(x) using polynomial representation (see Table 5). To construct effective potentials for Fokker–Planck equations for head monomers, we took the continuum model from the discrete values, which is the number of monomers in the (*z* > 0) region. What matters in an FP equation is the height of the barriers. A continuous free energy profile was made to catch the free energy height properly when the head monomer engaged toward the potential. Note that the initial position of the head was set to be at *m* = 4, where the head monomer was still confined by the pore. For *m* = 5, head monomer was released from the pore across the electric potential to the trans side. There was a small energy barrier against the negatively charged monomers toward the end of channel. (See Figure 2). The potential values at discrete points were obtained by taking the logarithm of the success rate increment (Equation (Equation 4)). There was discontinuity in our geometric environment between *m* = 4 and *m* = 5. Therefore, to capture the electrostatic properties of the channel potential, we used two polynomials before and after $x_{0}$ = 4.5 in the polynomial expression (Table 5). The polynomial representations of potentials for various values of $\epsilon_{\mathrm{HH}}$ are shown in Figure 8a. We wanted to describe the polymer under translocation through a Fokker–Planck equation for the motion of the head of the polymer. The translocation system is complex, and its friction depends on the translocated length and excited internal polymer modes. Strictly speaking, the polymer should be described by a (Fokker–Planck) equation in configurational space. The translocating PE is confined in the channel or collapsed into a globule. Except when a strongly fluctuating PE chain section is involved, such as in a good solvent ($\epsilon_{\mathrm{HH}}$ = 0.1) or theta solvent condition ($\epsilon_{\mathrm{HH}}$ = 0.3) outside the channel, it is legitimate to solve the simple Fokker–Planck equation for the head monomer, which ignores internal modes [34,35]. The Fokker–Planck equation with the proper potential is expected to be useful throughout.

The probability to find the head at *y* after time *t* starting from *x* at time t=0, p(y,t|x,0), satisfies the homogeneous Fokker–Planck equation:(5)$$\frac{\partial }{\partial t}p\left(y,t|x,0\right)=\frac{1}{\zeta }\frac{\partial }{\partial y}\left[\frac{dU(y)}{dy}p\left(y,t|x,0\right)\right]+D\frac{\partial^{2}}{\partial y^{2}}\left[p(y,t|x,0)\right].$$

Because we considered the translocation of the polymer, we set the boundary conditions at the ends of the interval inside, wherein the head of the polymer was constrained to m∈[a,b]. The boundary conditions for the translocation are absorbing boundaries at both *a* = 0 and *b* (=$m^{*}$ = 10), which means that, when the head of the polymer reaches the absorbing barrier, it is removed from the system so that the probabilities of being on the boundaries are zero.
(6)p(y=a,t|x,0)=p(y=b,t|x,0)=0.

Then, the probability, $\pi_{b}\left(x\right)$, of exit through *b* is given by Equation (Equation 4), and we find the mean exit time as follows:(7)$$\begin{array}{ccc} T_{1}\left(b,x\right) & = & \frac{1}{D\pi_{b}\left(x\right)\int_{a}^{b}e^{U(y)}dy}\left[\int_{a}^{x}e^{U(y)}dy\cdot \int_{x}^{b}e^{U(y^{\prime })}dy^{\prime }\int_{a}^{y^{\prime }}e^{-U(z)}\pi_{b}\left(z\right)dz\right.\\ & & -\left.\int_{x}^{b}e^{U(y)}dy\cdot \int_{a}^{x}e^{U(y^{\prime })}dy^{\prime }\int_{a}^{y^{\prime }}e^{-U(z)}\pi_{b}\left(z\right)dz\right]. \end{array}$$

The detailed derivation for the translocation times (including higher moments) for the head of the polymer by the Fokker–Planck equation is shown in Appendix C (See also Ref. [36]).

A complete picture was also obtained numerically using the Runge–Kutta method, and the results of $\pi_{b}$ and $T_{1}$ were consistent with the integral expressions of Equation (Equation 4) and Equation (Equation 7). Figure 8 and Table 6 summarize the success rates $\pi_{b}$ and the exit times $T_{1}$ obtained from Equations (Equation 4) and (Equation 7), with absorbing boundaries at *a* = 0 and $b=m^{*}$ = 10 and with the starting position at *x* = 4. In order to compare with the simulation results, the time unit $t_{0}=1/D=\zeta /k_{B}T$ ($k_{B}T$ = 1) was multiplied. The results were compared with the waiting time $t_{w}$ obtained from simulations, where we set the initial head position at the *trans* end (i.e, *m* = 4) and the exit boundary to be *m* = 0 (rejected) and *m* = 10 ($n_{t}$ = 5). The two results agreed well for $\epsilon_{\mathrm{HH}}\leq 0.7$ and started deviating from each other at $\epsilon_{\mathrm{HH}}=0.9$.

## 4. Conclusions

We studied the translocation of (uni)globular polyelectrolytes by means of MD simulations. The uniglobular structure is expected for hydrophobic PEs at high salt conditions, which are typical for translocation through a pore driven by an electric field. The PE was simulated in the HP model, where neutral monomers are hydrophobic and charged ones are hydrophilic. The hydrophobicity of the H monomers sets the density of the PE globule. The main new feature with respect to our (and others’) previous work is the unraveling of the PE globule during (prior to) translocation.

In this work, we used simple periodic sequences in which extra barriers against translocation of antagonistic charges, as described previously for polyampholytes [15], were not present. Hydrophobicity has several effects. More hydrophobic globules are harder to unwind and present higher monomeric friction during unwinding due to their increased density. The friction effect is especially important at high hydrophobicity, where the globules may become quasi-glassy [33], despite their mesoscopic size. Translocation processes can be decomposed into three stages: capture, waiting, and drift. We only studied the two latest stages and started with a PE well engaged into the pore. The waiting process extends as long as the PE still fluctuates back to its initial position. The waiting time increases with hydrophobicity. At a high hydrophobicity it involves the nucleation of a trans globule, which manifests as an extra (secondary) barrier against translocation (Figure 8a). During the drift process, the translocated mass increases nearly linearly with time. The drift time is found to go through a shallow minimum, which is also visible in the total time (waiting time + drift time), for a weak hydrophobicity before increasing markedly for a larger hydrophobicity. The friction due to the channel potential prevails in weak hydrophobic regimes. The channel potential is heterogeneous in small scales, and this causes extra dissipation.

The polymer globules studied here were reminiscent of globular proteins, even though proteins typically carry the charges of both signs. It is currently assumed/accepted that, in some processes, the proteins could be partially denatured and reach the softer polymer globule state either dry or wet [22]. The wet globule has similar properties to the HP globules studied here. It would be interesting to investigate the translocation of globular proteins with well-controlled denaturation.

For real proteins, the side chains of the amino-acids would affect translocation behavior. Although it depends on the size of channel, the translocation times showed non-monotonic increases with the number of arms [37,38]. The study for the translocation of hydrophobic chains with side-groups will be investigated in the near future.

## Figures and Tables

**Figure 1 polymers-15-02550-f001:**
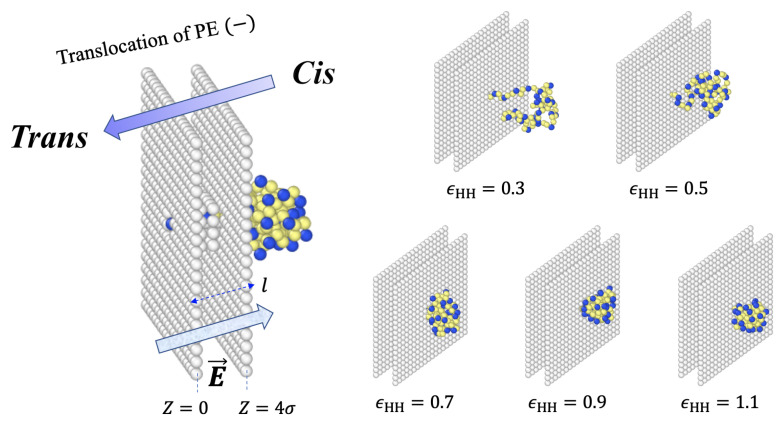
Schematics of the translocation process. A bilayer membrane of thickness l=5σ separates the *cis* and *trans* regions. Negatively charged polymers engage through the pore and are driven by electric field from the *cis* side to the *trans* side. Snapshots show conformations of captured PEs at various solvent conditions. Blue colored monomers carry charge and are considered as hydrophilic (polar). Yellow colored monomers are electrically neutral and are hydrophobic. Hydrophobic monomers interact through an attractive Lennard–Jones potential of strength $\epsilon_{\mathrm{HH}}$.

**Figure 2 polymers-15-02550-f002:**
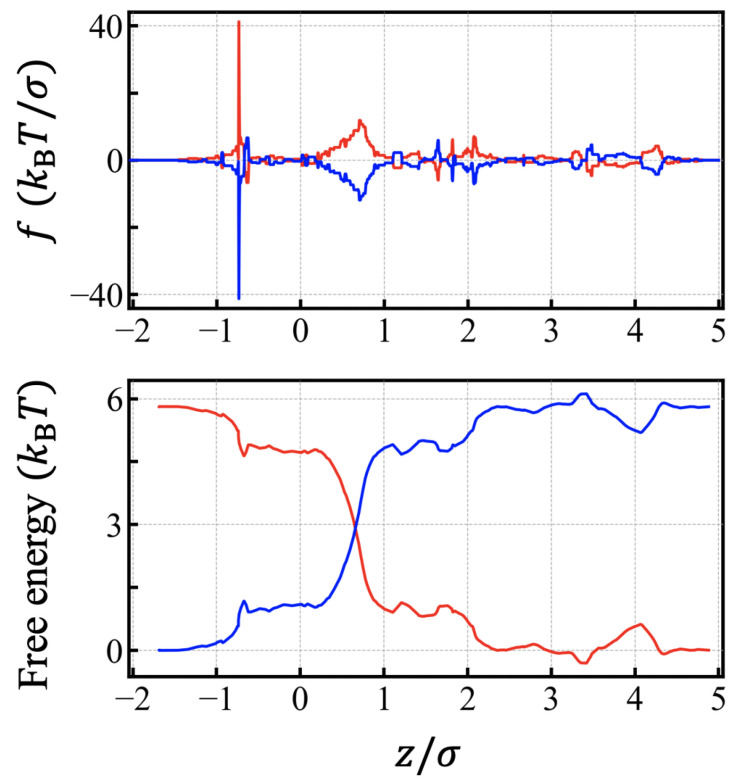
The force field (**top**) and the free energy (**bottom**) of electrostatic origin when external electric potential 150 mV is applied across α-hemolysin pore. Total potential drop is ∼$5.8\;k_{B}T$ over the distance of 6σ (=12 nm). The unit of force $k_{B}T/\sigma$ corresponds to ∼2 pN in real units. Forces also reflect the fixed charge arrangement inside the channel. Blue lines are for anions and red lines for cations.

**Figure 3 polymers-15-02550-f003:**
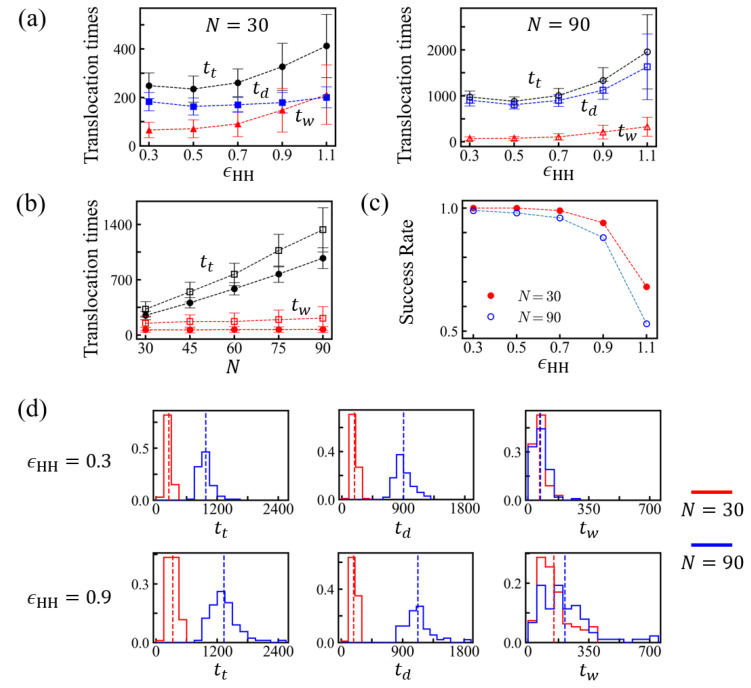
(**a**) Total translocation time $t_{t}$, drift time $t_{d}$, and waiting time $t_{w}$ of PEs for *N* = 30 (left) and *N* = 90 (right) at various solvent conditions. (**b**) Translocation times $t_{t}$ (black) and waiting times $t_{w}$ (red) as a function of chain length *N* for $\epsilon_{\mathrm{HH}}$ = 0.3 (•) and $\epsilon_{\mathrm{HH}}$ = 0.9 (□). (**c**) Probabilities of successful translocation at various $\epsilon_{\mathrm{HH}}$ for *N* = 30 and *N* = 90. (**d**) Distributions of total translocation times $t_{t}$, drift times $t_{d}$, and waiting times $t_{w}$ measured for *N* = 30 and *N* = 90 with $\epsilon_{\mathrm{HH}}$ = 0.3 and 0.9, respectively. The mean values are indicated as dotted lines.

**Figure 4 polymers-15-02550-f004:**
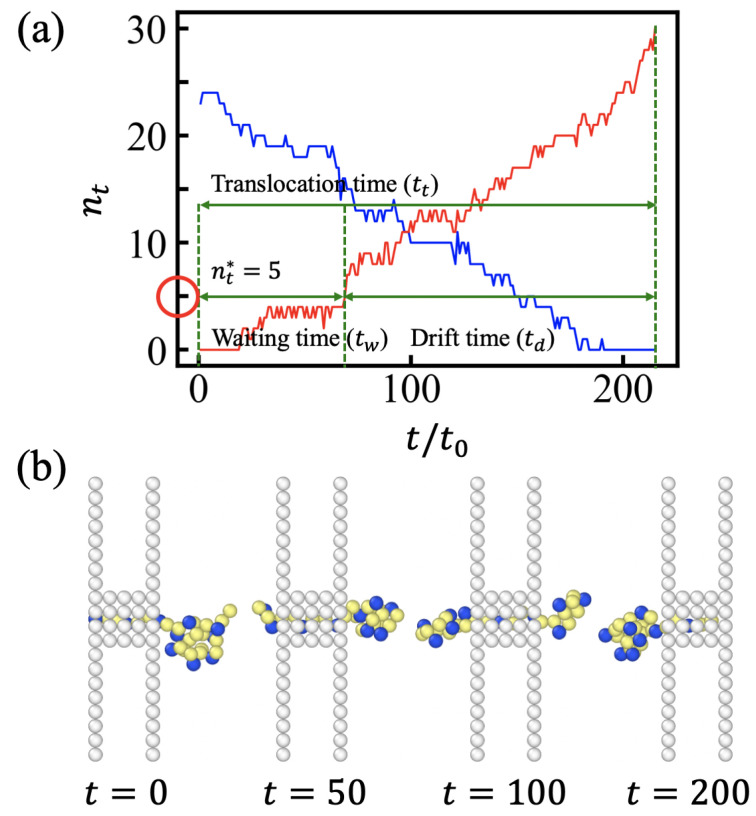
(**a**) The number of monomers in the *cis* side (blue) and the *trans* side (red) in a simulation run ($\epsilon_{\mathrm{HH}}$ = 0.7 and *N* = 30). The waiting time $t_{w}$ is the MD time required for the number of *trans* side monomers to be $n_{t}^{*}$ = 5. After $t_{w}$, the time until the tail of the chain leaves the channel is the drift time $t_{d}$. Representative conformations are shown in (**b**). From left to right, the figures show an initial conformation of PE at gate, a conformation in waiting period and in drift regime, and a conformation almost translocated. Yellow and blue monomers represent H- and P-type monomers, respectively.

**Figure 5 polymers-15-02550-f005:**
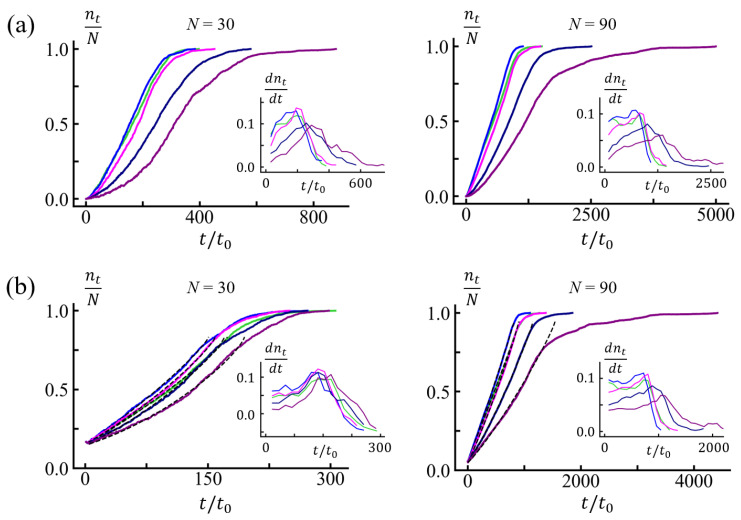
(**a**) The number of monomers $n_{t}\left(t\right)$ in the *trans* side for *N* = 30 (left) and *N* = 90 (right). Insets show translocation speed $\frac{dn_{t}\left(t\right)}{dt}$. (**b**) The number of monomers in the *trans* side and the translocation speed (insets) during drift period for *N* = 30 (left) and *N* = 90 (right). Time $t_{w}$ was reset to *t* = 0 ($t>t_{w}$). For all panels, the green, blue, magenta, navy, and purple lines are for $\epsilon_{\mathrm{HH}}$ = 0.3, 0.5, 0.7, 0.9, and 1.1, respectively. Each curve was obtained by averaging over 100 trials. Note that translocation was fastest at $\epsilon_{\mathrm{HH}}$ = 0.5 (blue).

**Figure 6 polymers-15-02550-f006:**
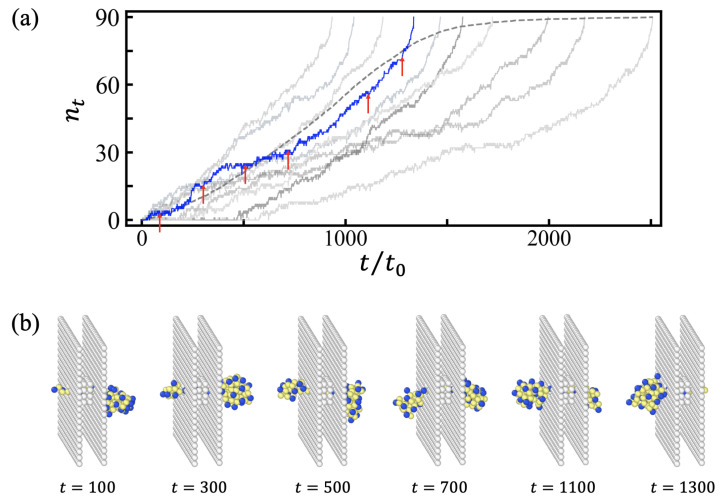
(**a**) Several typical trajectories (grey lines) of $n_{t}\left(t\right)$ for *N* = 90 with $\epsilon_{\mathrm{HH}}$ = 0.9. The average trajectory is shown as a dashed line for comparison. (**b**) Conformations of polyelectolytes during the translocation. The corresponding trajectory is shown as a blue line and the times corresponding to the snapshots are indicated by arrows. Translocation process appeared to be paused when no monomer was released from the *cis* side globule. Yellow and blue colored spheres represent hydrophobic and hydrophilic charged monomers, respectively.

**Figure 7 polymers-15-02550-f007:**
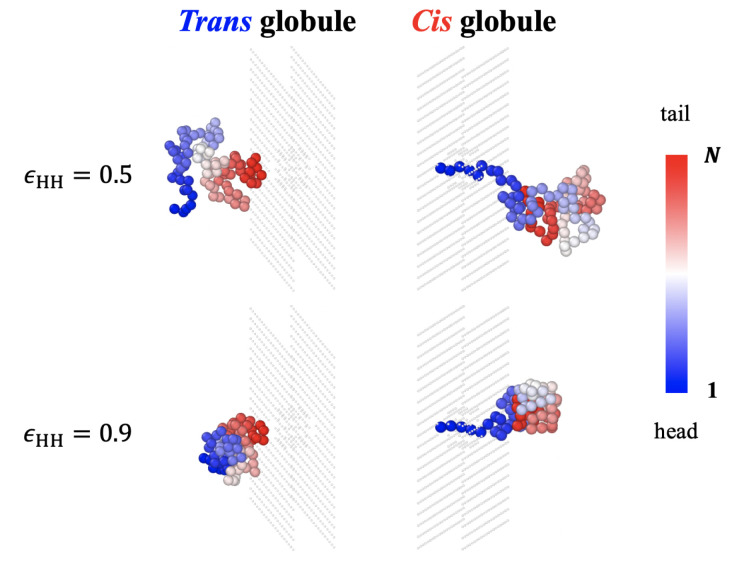
Illustrations of monomer mixing in *cis* and *trans* sides with *N* = 90 for $\epsilon_{\mathrm{HH}}$ = 0.5 and 0.9. Colors of monomers indicate the order of translocation from head to tail.

**Figure 8 polymers-15-02550-f008:**
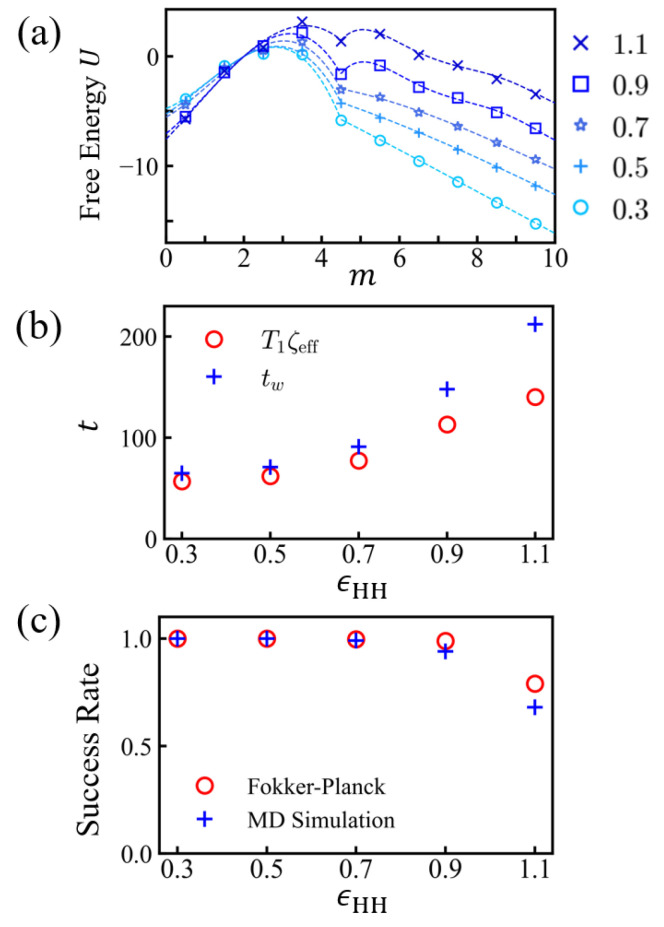
(**a**) Effective potential U(m) of ${\left(100\right)}_{10}$ sequence (*N* = 30) for various values of $\epsilon_{\mathrm{HH}}$. *m* is the number of monomers passing the *cis*-side boundary of the membrane. The continuous forms U(x) (dashed lines) was obtained by fitting discrete values of U(m) with polynomials, as in Table 5. (**b**) Estimates of exit time $T_{1}$ and (**c**) success rate using the Fokker–Planck equation (Table 6) with absorbing boundaries at *m* = 0 and *m* = 10 ($n_{t}$ = 5). The waiting times $t_{w}$ from simulation are shown for comparison.

**Table 1 polymers-15-02550-t001:** Translocation times of PE (a) *N* = 30 and (b) *N* = 90 with various values of $\epsilon_{\mathrm{HH}}$. $\epsilon_{\mathrm{HH}}$ = 0.1 and 0.3 correspond to good and theta solvent conditions, respectively.

(a) *N* = 30				
$\epsilon_{\mathrm{HH}}$	success rate	$t_{t}$	$t_{w}$	$t_{d}$
0.1	1.00	260 ± 50	71 ± 29	189 ± 39
0.3	1.00	251 ± 41	65 ± 28	187 ± 39
0.5	1.00	235 ± 54	71 ± 37	163 ± 35
0.7	0.99	261 ± 57	91 ± 52	170 ± 31
0.9	0.94	327 ± 97	148 ± 90	179 ± 42
1.1	0.68	413 ± 130	212 ± 123	201 ± 43
(b) *N* = 90				
$\epsilon_{\mathrm{HH}}$	success rate	$t_{t}$	$t_{w}$	$t_{d}$
0.1	1.00	1035 ± 144	69 ± 35	966 ± 134
0.3	0.99	974 ± 132	71 ± 38	903 ± 124
0.5	0.98	881 ± 100	76 ± 42	805 ± 91
0.7	0.96	1006 ± 157	106 ± 69	900 ± 130
0.9	0.88	1333 ± 280	212 ± 149	1121 ± 195
1.1	0.53	1959 ± 810	328 ± 207	1631 ± 714

**Table 2 polymers-15-02550-t002:** The effective frictional constant obtained from early (drift) regime by the slope ∼$dn_{t}\left(t\right)/dt$ assuming the constant force. The number in the parenthesis is from the average slope for times $n_{t}<N/2$. Fitting results, $k^{\prime }$ and ζ, are obtained according to Equation (Equation 2) in drift regime ($t>t_{w}$).

$\epsilon_{\mathrm{HH}}$	*N* = 30			*N* = 90		
	$\zeta^{\mathrm{eff}}$	$k^{\prime }$	ζ	$\zeta^{\mathrm{eff}}$	$k^{\prime }$	ζ
0.3	20.0 (21.6)	2.00	18.1	23.6 (24.1)	0.2	23.0
0.5	17.4 (20.3)	1.90	15.8	21.4 (22.4)	0.5	20.5
0.7	18.8 (24.3)	2.50	16.7	25.2 (27.2)	1.7	22.5
0.9	20.6 (32.6)	3.20	17.5	32.8 (38.3)	2.1	28.5
1.1	24.9 (41.3)	4.40	19.6	45.1 (53.0)	2.2	38.0

**Table 3 polymers-15-02550-t003:** Translocation time dependence on chain length *N* for $\epsilon_{\mathrm{HH}}$ = 0.9.

*N*	Success Rate	$t_{t}$	$t_{w}$	$t_{d}$
30	0.94	327 ± 97	148 ± 90	179 ± 42
45	0.96	546 ±125	171 ± 84	376 ± 89
60	0.85	769 ± 142	171 ± 110	598 ± 111
75	0.83	1070 ± 205	195 ± 121	876 ± 168
90	0.88	1333 ± 280	212 ± 149	1121 ± 195

**Table 4 polymers-15-02550-t004:** Translocation times of sequences (100)${}_{10}$, (010)${}_{10}$, and (001)${}_{10}$. The corresponding probabilities of success are shown in parenthesis below the translocation times.

	0.3	0.5	0.7	0.9	1.1
(100)	252	235	261	327	413
	(1)	(1)	(0.99)	(0.94)	(0.68)
(010)	232	225	244	263	338
	(0.88)	(0.89)	(0.78)	(0.79)	(0.57)
(001)	241	241	220	210	375
	(0.29)	(0.29)	(0.18)	(0.03)	(0.02)

**Table 5 polymers-15-02550-t005:** Polynomial representation of free energies: Region 1 [0:4.5] defined as $a_{3}x^{3}+a_{2}x^{2}+a_{1}x+a_{0}$. Region 2 [4.5:10] defined as $b_{5}{\left(x-x_{0}\right)}^{5}+b_{4}{\left(x-x_{0}\right)}^{4}+b_{3}{\left(x-x_{0}\right)}^{3}+b_{2}{\left(x-x_{0}\right)}^{2}+b_{1}\left(x-x_{0}\right)+b_{0}$ with $x_{0}$ = 4.5. The drift regime curves were reconstructed by using the fitted value of $k^{\prime }$ and electrical field slope $E_{0}$ = −1.9.

$\epsilon_{\mathrm{HH}}$	$a_{3}$	$a_{2}$	$a_{1}$	$a_{0}$	$b_{5}$	$b_{4}$	$b_{3}$	$b_{2}$	$b_{1}$	$b_{0}$
0.3	−0.336	1.143	1.437	−4.758	0	0	0.0022	−0.025	−1.819	−5.742
0.5	−0.251	0.676	2.275	−5,246	0	0	0.003	−0.070	−1.232	−4.194
0.7	−0.287	0.986	1.939	−5.526	0	0	0.011	−0.174	−0.705	−2.959
0.9	−0.282	0.908	2.831	−7.038	0.0202	−0.3336	1.9913	−5.1950	4.2521	−1.565
1.1	−0.160	0.333	3.753	−7.558	0.0196	−0.3228	1.9213	−4.9750	3.9750	1.479

**Table 6 polymers-15-02550-t006:** The exit times $T_{1}$ and success rate $\pi_{b}$ obtained from Equations (Equation 4) and (Equation 7). In order to compare with simulation results $t_{w}$, time unit $\zeta^{\mathrm{eff}}$ was multiplied.

$\epsilon_{\mathrm{HH}}$	$\pi_{b}$	$T_{1}$	$T_{1}\zeta^{\mathrm{eff}}$	$t_{w}$
0.3	1.000	2.845	56.9	65
0.5	0.999	3.561	62.0	71
0.7	0.997	4.113	77.3	91
0.9	0.989	5.488	113.1	148
1.1	0.615	5.634	140.3	212

## Data Availability

Not applicable.

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
