# Peer review of "Translocation of Hydrophobic Polyelectrolytes under Electrical Field: Molecular Dynamics Study"

_polymers, 2023, doi:10.3390/polym15112550_

Round 1
Reviewer 1 Report
In this work, the authors investigated translocation dynamics of a polyelectrolyte chain driven under different solvent conditions. They studied translocation time and success rate, and calculated effective friction coefficient by means of molecular dynamics simulations. The free energy profile was then studied and the results were compared with the ones solved from a simplified Fokker-Planck equation. The studied topics and results look intriguing. However, the presentation is not very clear. The analyzed subjects can go deeper. A major revision is requested in order to fulfil the standard of publication. Followings are some issues to be clarified.
1. Which kind of boundary condition is used in the simulations? In Line 88, the authors describe “we use the LJ potential without cut-off length”. How is the force calculation being implemented in the study?
2. The strength of the driving electric field should be given explicitly in the model section. The authors said “the potential difference between the cis and trans regions is ~5.8kT” (in Line 112). I do not understand where comes the uncertainty in the potential difference because the pore region has been clearly defined (-1<z<5) in the simulations?
3. How did the authors have the Figure 2? It cannot be found in Ref. 14. The authors should tell how they extracted the data and explain clearly from which case of study or figure in Ref. 14 the data were adopted.
4. The threshold to distinguish the waiting time and the drift time is set at nt=5 in the study, because “The chain dose not retreat back, once the number of translocation monomers nt reach[es] 5 (in Line 153)”.
Shouldn’t the threshold depend on the chain length because the retreating force from the cis side increases with the chain length? Similarly, the threshold should also depend on the solvent condition. Therefore, the choice “nt=5” looks arbitrary. The reported waiting time and drift time depend directly on the choice. It may have unexpected impact on the conclusions if the threshold is not accurate. The authors should find a rigorous way to define the threshold.
5. The authors regarded the effective friction coefficient as a constant in Table 2. It lacks justification because the curves nt shown in Figure 5 obviously do not have a constant slope. The authors should show time derivative of nt to justify the assumption.
Eq.(2) in Page 7 has just said the contradiction. The effective friction coefficient (fe/(d nt/d t)) deduced from Eq(2) is obviously a function of nt and thus time-dependent.
The dependence of the effective friction coefficient on the translocation coordinate has been pointed out and studied for a general ejection and translocation process in the papers: Phys. Rev. Lett. 123 (2019) 267801 and Scientific Reports 11 (2021) 14721.
6. The free energy profile in the early stage (m<=10) of translocation was calculated in Figure 7, and then fit by two polynomials in Table 5. The fitting looks not rigorous because the number of data points (10) on a curve is equal to the number of the fitting parameters. Also a cusp displays at m=4.5. Without proper explanation, the choice x0=4.5 in the fitting polynomial looks arbitrary.
7. The density correlation studied in Appendix A presents only the distributions of the two type of monomers at an initial state, prior to the translocation. It will be interesting if the authors could study the distributions of the monomers during and at the end of the translocation. Particularly, the chain forms two globules on the cis and trans sides, as shown in Figure 6. Currently, the studied subject is not deep enough. The effective friction coefficient is expectedly related to the globule structures on the two sides of the pore during the translocation; refer to the papers: Polymers 10 (2018) 1129 and ACS Omega 5 (2020) 19805.
Reviewer 2 Report
In this work, the authors studied the translocation of hydrophobic PE chains via MD simulations under an applied potential. This is an interesting work which helps to fundementally understand the friction and diffusion of the system. The below issues should be clarified before it is accepted.
1. The title is too general and needs to be further clarified to indicate that only the MD simulations are used with an applied electric field.
2. In the first Langevin equation, why the authors assume a constant friction coefficient in an uneven system? The friction coefficient should depends on the collision of the surrounding molecules which is usually not a constant value, which can be solved by the MD simulations. This part needs to be further discussed.
3. It is very interesting to see the contribution of the channel friction and the internal friction respectively. Is the internal friction a dominating role here?
4. Velocity verlet and Runge-Kutta methods are used respectively in different numerical calculations. Why and what is the difference?
5. It seems that the PE chain is a string of particles without detailed molecular structures. This maybe not suitable for describing the real polymers, where the size and types of side groups are expected to have big effect. This part at least should be discussed.
6. Why a εHH = 0 as a background is not simulated? What would the authors expect for the results in this case?
Minor editing of English language required
Reviewer 3 Report
My comments are provided in the attached pdf.

Round 2
Reviewer 1 Report
The authors have made the presentation clearer and added more explanations, figures, and related references in this resubmittal. Most of the issues raised by me have been resolved. However, two problems related to molecular dynamics simulations should be further clarified.
(1) In the response letter, the authors said: “We put the membrane and a translocating chain in an infinite space. Hence no boundary is imposed. … Typically for N = 30, 20? × 20? is sufficient. For N = 90, especially in good solvent condition, it needs to cover the area of 40? × 40?.”
It looks that the simulation box has infinite size. So no boundary condition, such as the conventional periodic boundary condition, is set in this study. In the simulation box, the authors placed a piece of membrane of finite dimension, varied from 20?×20?×5? to 40?×40?×5?. The chosen lateral dimension of the membrane depends on the chain length to guarantees that the chain will not span across the membrane from the “outer periphery” in a good solvent condition.
I learned the above information totally from the response. I think it is important to describe the details explicitly in the manuscript so that anyone by reading the manuscript can have a clear picture about the setup of the simulations.
(2) For the second question, the authors answered: “We used the force field of which the integrated value over the interval −1? < ? < 5? corresponds to the potential difference 125 meV. The uncertainty in the text is originated from the unit conversion. When we convert the electric potential difference 125 meV to the thermal unit ??, it is approximated to be 5.8 ?? assuming room temperature.”
I still do not know the exact value of the force strength (in reduced unit) applied inside the pore to drive a P-type monomer. The simulations were performed by using reduced unit. The authors do not mention anything about how they converted the reduced unit to a real unit. For example, what is the length (in nm) corresponding to 1 ? in the simulation? I think it is important to provide the detail setting for the force strength because the behavior of a forced translocation depends on the driving force.
Reviewer 2 Report
I thank the authors for their efforts, and I believe this work can be published now.
Author Response
We thank the referee 2 for supporting the publication of our manuscript.